# Swarming Transition in Super-Diffusive Self-Propelled Particles

**DOI:** 10.3390/e25050817

**Published:** 2023-05-18

**Authors:** Morteza Nattagh Najafi, Rafe Md. Abu Zayed, Seyed Amin Nabavizadeh

**Affiliations:** Department of Mechanical Engineering, University of Akron, Akron, OH 44325, USA

**Keywords:** super-diffusive Vicsek model, Levy flights, second-order phase transition, critical exponents, 05., 05.20.-y, 05.10.Ln, 05.45.Df

## Abstract

A super-diffusive Vicsek model is introduced in this paper that incorporates Levy flights with exponent α. The inclusion of this feature leads to an increase in the fluctuations of the order parameter, ultimately resulting in the disorder phase becoming more dominant as α increases. The study finds that for α values close to two, the order–disorder transition is of the first order, while for small enough values of α, it shows degrees of similarities with the second-order phase transitions. The article formulates a mean field theory based on the growth of the swarmed clusters that accounts for the decrease in the transition point as α increases. The simulation results show that the order parameter exponent β, correlation length exponent ν, and susceptibility exponent γ remain constant when α is altered, satisfying a hyperscaling relation. The same happens for the mass fractal dimension, information dimension, and correlation dimension when α is far from two. The study reveals that the fractal dimension of the external perimeter of connected self-similar clusters conforms to the fractal dimension of Fortuin–Kasteleyn clusters of the two-dimensional Q=2 Potts (Ising) model. The critical exponents linked to the distribution function of global observables vary when α changes.

## 1. Introduction

There has been a significant increase in attention towards emergent behaviors, particularly collective motions in active systems [1]. These systems exhibit large-scale cooperative phenomena and sometimes scale-invariant patterns, resembling the standard theory of statistical mechanics and critical phenomena. To draw a connection, the concept of “phase” is assigned to each collective mode of behavior of agents, and the term “phase transition” is used. Swarming is an example that is observed in various biological systems, e.g., fish schools with milling dynamics [2], bacterial systems [3,4,5,6], biofilm formation [7], and marching locusts [8]. Observations of self-organized critical behavior in systems such as midge swarms [9] prompt the question of what the significance of criticality is in these systems. Among the mathematical models [10], the Vicsek model (VM) [11] serves as a minimal prototypical example of active systems with self-propelled constituents that show swarming transition.

The VM is a simple yet extensively studied model due to its ability to exhibit new features and diverse modes of collective behavior. For example, it was mapped onto complex networks [12] and XY spin chains [13,14,15]. Other studies have focused on the impact of angular noise [16] and hierarchical societies [17], with the aim of uncovering the key physical factors that contribute to emergent behaviors and its response to additional interactions/physical parameters; see [18].

Initial studies asserted that the VM exhibits critical behavior at the transition point [11,13,14,15], but later research showed that the transition is actually of the first order [19,20]. This discrepancy was attributed to the small finite size effects of the system. However, other studies suggest that the order of the transition may depend on how noise is added to the system [21] or could be an artifact of finite size and boundary conditions [22]. This leaves an important unanswered question: How do the long-range interactions [9], which lead to criticality in many systems [23,24,25], impact the collective motion when these individuals are swarming together?

A recent study [26] found that the lack of fluctuations, or complete synchronization with the flock, in the VM and other classical models, such as the Cucker–Smale model, can reduce the adaptive response of a flock, which may be undesirable in certain scenarios. To increase the adaptive sensitivity to external threats, criticality and correspondingly large fluctuations are essential, as information corresponding to a local perturbation moves faster due to the scale-free correlations [27,28]. It is now well known that criticality is a vital ingredient for a flock state to survive in the presence of external threats [29,30].

We argue that the presence of scale-free *stochasticity* in, for instance, the flight distance of the constituents is a crucial factor for achieving criticality, which is absent in classical counterparts. To establish a closer connection with real-world scenarios, it is essential to incorporate scale-free stochasticity that generates anomalous diffusion, which is frequently observed in active systems. Examples include Levy flights of wandering albatrosses [31] and other animals [32], super-diffusive intracellular transport [33], entangled F-actin networks [34], microtubule-associated motors within a living eukaryotic cell [35], living yeast cells [36], mRNA molecules inside live E. coli cells [37,38], telomeres in the nucleus of mammalian cells [39], biomolecules in solution and living cells [40], the pathway of an Adeno-associated virus [41], and epithelial cell migration (with Levy flights as the asymptotics of the *q*-Weibull distributions) [42]. In most cases, super-diffusion is observed, which enhances the exploration process (e.g., foraging for animals) [31,32]. In this paper, we investigate the impact of introducing Levy flights into the system dynamics, resulting in super-diffusive stochasticity. This leads to the criticality of the system for sufficiently small step exponents (α) of Levy flights.

The paper is organized as follows: In the next section we describe our model. Section 3 presents the simulation results, while Section 4 presents the mean field results. Section 5 investigates the geometrical and global features of the model and is divided into two subsections. The first subsection, Section 5.1, describes the mass fractal dimension and higher-order dimensions of the density field. In the second subsection, Section 5.2, we present our contour line analysis for the density field. We conclude the paper with a summary of our findings in the final section.

## 2. The Model

In the ordinary VM, the agents (labeled by i∈[1,N], where $N=\rho L^{2}$ represents the total number of active particles, ρ is the density of the particles and *L* is linear size of the system) undergo correlated random walks in the system with an interaction range *R* such that any active particle inside a disk of radius *R* is fully *seen* by the central particle. The time evolution of the position of the *i*th particle at time *t* ($x_{i}\left(t\right)$) is given by
(1)$$\begin{array}{cc} \; & x_{i}\left(t+\Delta t\right)=x_{i}\left(t\right)+v_{i}\left(t\right)\Delta t\\ \; & \theta_{i}\left(t+\Delta t\right)={\left.\langle \theta_{t}\rangle \right|}_{\left\{R_{i}\right\}}+\eta \zeta_{i} \end{array}$$
where $\theta_{i}\left(t\right)$ is the direction of motion of the *i*th particle at time *t*, $v_{i}\left(t\right)=v_{i}\left(\cos\theta_{i}\left(t\right),\sin\theta_{i}\left(t\right)\right)$, $v_{i}\Delta t$ is the distance that the *i*th particle traverses in the time interval [t,t+Δt], $\left\{R_{i}\right\}$ shows the set of particles at a distance less than or equal to *R* from the *i*th particle, and ${\left.\langle \theta_{t}\rangle \right|}_{\left\{R_{i}\right\}}\equiv \mathrm{Arg}\left[\sum_{j\in \left\{R_{i}\right\}}e^{\theta_{j}\left(t\right)}\right]$. $\zeta_{i}$ is a uniform random number in the interval $\left[-\frac{1}{2},\frac{1}{2}\right]$, and η is the strength of the disorder.

In the ordinary VM, where the particles have a constant velocity $v_{i}^{\mathrm{VM}}=v_{0}$ (∀i∈[1,N]), the coherence order parameter defined as $\phi_{\eta }\left(t\right)\equiv \left|\frac{1}{N}\sum_{i=1}^{N}v_{i}\left(t\right)\right|$ is zero (non-zero) in the disordered (ordered) phase. This indicates that the particles move in a spatiotemporally coherent (incoherent) fashion [11], and provide information about the degree of orientation of the particles’ motion. There is a transition point $\eta_{c}$, above (below) which $\phi \left(\eta \right)\equiv {\langle \phi_{\eta }\left(t\right)\rangle }_{t}$ and the systems are in the disordered (ordered) phase. ${\langle \rangle }_{t}$ is defined as the time average over a considerable time interval. In our model, we let the particles’ velocity obey the Levy flight distribution:(2)$$p_{\mathrm{Levy}}\left(v\right)\propto \frac{1}{v^{\alpha +1}}\Theta \left(v_{\max}-v\right),$$
where α is a “step index” which generates correlations, and Θ(x) is a step function (Θ(x)=1 for x≥0, and zero otherwise).

The occurrence of rare events (long-range) flights in the process introduces correlations by allowing the particles to access and persist in specific regions of space. Consequently, correlations in the particles’ positions over time emerge [43]. Θ(x) is considered in our model to prevent unphysical rare events, such as flights above the threshold speed $v_{\max}$ imposed by physical conditions ($l_{\max}\equiv v_{\max}\Delta t$), which serves as an IR cut off in the problem. In the analytical calculations, we require also a UV cutoff ϵ to ensure that the average flight ${\bar{l}}_{\alpha }\equiv \int_{\epsilon }^{l_{\max}}lp_{\mathrm{Levy}}\left(l\right)dl=\frac{A}{\alpha -1}\left(\epsilon^{1-\alpha }-l_{\max}^{1-\alpha }\right)$ is well-defined, where $A^{-1}\equiv \frac{1}{\alpha }\left(\epsilon^{-\alpha }-l_{\max}^{-\alpha }\right)$ is a normalization constant. It is customary to consider this cutoff as the smallest scale in the problem, e.g., a lattice constant in the models on the lattice. We set it to *R*, which is the lattice constant in our setup. For 1<α<2, the average length and its variance diverge as the limits of ϵ→0 or $l_{\max}\to \infty$ are approached, which results in the central limit theorem being invalid. In this situation, the distribution of the Levy random walkers is described by α-stable distributions as reported in [44]. We refer to the particles in this regime as super-diffusive active particles because the diffusion exponent is $\frac{\alpha }{2}>0.5$ [45].

For the ordinary VM, there is a coexistence of ordered and disordered phases at the transition point (which is of the first order). This leads to the existence of well-defined phase boundaries, known as phase coexistence [17], and results in a bimodal distribution function for the order parameter. The bimodal distribution can be either spatial or temporal, indicating that the two phases may either spatially coexist or dominate during different time intervals [17,46]. The existence of a bimodal distribution function may directly lead to a gap for ϕ(η) at the transition point [47], around which the hysteresis effect is observed. In this state, the system stays during most of the observation time in the vicinity of one peak of p(ϕ) or, in other words, a metastable branch is observed. A hysteresis loop results from the system’s resistance to entering the new peak [17]. The gap is defined as the difference between these two peak points exactly at the transition point. This is in accordance with the discontinuous (first order) phase transitions, where the phases coexist with well-defined boundaries. However, with the second-order phase transitions, the system shows self-similar patterns at the transition point, making it challenging to attribute a specific phase to a part of the system [47]. In our model, the scale-free stochasticity induces the evaporation of the two phases by promoting “tunneling” between the peaks. The amplitude of this effect is determined by the value of α.

## 3. Simulation Results

We simulated the system for $\frac{L}{R}=32,64,128$ and 256, α∈[0.8–1.95], and ρ=2. Additionally, *R* and Δt were set to 1. The active particles were initially distributed randomly in the system with random uncorrelated movement orientations. For the systems that exhibit a hysteresis effect, we moved forward (increase η) and backward (decrease η), and for both cases, we changed η by $\delta \eta =3\times 10^{-6}$ in each time step. To control statistical fluctuations, we generated three samples at each time step. We used the maximum likelihood estimation (MLE) method [48] to estimate the best values for the exponent and corresponding error bars. For the data collapse analysis, we selected the graph of the largest system size as a base. We calculated the $\chi^{2}$ value for all other graphs by measuring the cumulative distance of the points from this base graph. The distribution of $\chi^{2}$ values was then used to determine the best exponents and corresponding error bars. This technique was used to better understand the relationship between different variables in the system. The density pattern of the model in the transition point is strongly dependent on the value of α as shown in Figure 1.

Figure 1 investigates two different scenarios of a physical system with different values of α. In the first case, α=1.5, corresponding to Figure 1a, the system displays a self-similar pattern without any phase separation. The second case, where α=1.95, corresponding to Figure 1b, exhibits a strip-like dense (ordered) phase in a background of a dilute (disordered) phase. The self-similar pattern for α=1.5 is reminiscent of the second-order transition, while the phase separation for α=1.95 is a fingerprint of the first-order transition.

To quantify the order of the transition, we show the time series of ϕ and the corresponding probability distribution p(ϕ) in the Figure 2a for $\eta >\eta_{c}$, $\eta =\eta_{c}$ and $\eta <\eta_{c}$ for α=1.5 (left, with a single peak) and α=1.95 (right, with a bi-modal structure). The single peak structure was observed for all small enough α values, suggesting that the transition in our model for these α values does not follow the first-order phase transition paradigm. In contrast, the transition is *discontinuous* for α values around two. In Figure 2b, the ϕ-η plot for different α values demonstrates that increasing α shifts the graphs to the left, suggesting that the disordered phase is stabilized by higher α values. The Binder cumulant method was used to extract the transition points. This method is especially an effective tool for identifying the type of transition, as it is a continuous quantity for second-order phase transitions. It is defined as
(3)$$G_{\eta }=1-\frac{{\langle \phi_{\eta }^{4}\rangle }_{t}}{3{\langle \phi_{\eta }^{2}\rangle }_{t}^{2}}.$$

This function shows a sudden deep minimum at the transition point, which is characteristic of first-order phase transitions (upper inset of Figure 2b for α=1.5). This function is inspected further in Figure A1a in Appendix A. There, it is shown that the depth of the valley decreases as α decreases, which reveals the system crossover to regimes with properties similar to second-order phase transitions in sufficiently small α values. The observation is consistent with the modal structure of the distribution function, Figure A1b (see Appendix A for more details). While $G_{\eta }$ is theoretically expected to be *L*-independent at the second-order phase transitions [49], statistical uncertainties prevent the points from precisely intersecting at a single point. The method proposed in [50] is utilized to estimate $\eta_{c}$ in the thermodynamic limit. The method relies on identifying the intersection point ($\eta_{c}\left(L,L^{\prime }\right)$) of two graphs corresponding to two successive *L*s and extrapolating the obtained point to $L,L^{\prime }\to \infty$. The resulting transition point $\eta_{c}$ is shown in the lower inset of Figure 2b, exhibiting a decreasing behavior in terms of α, i.e., the disordered phase dominates as α increases.

Although the observations mentioned suggest similarities with second-order phase transition for small enough α values, additional statistical evidence is required to validate this hypothesis. According to the standard theory of second-order (continuous) phase transitions, the order parameter and the order parameter fluctuation $\chi_{\phi }\equiv L^{2}\left[\langle \phi^{2}\rangle -{\langle \phi \rangle }^{2}\right]$ satisfy the finite size scaling hypothesis [51]:(4)$$\phi_{\eta }=L^{-\beta /\nu }F_{\phi }\left(\epsilon L^{1/\nu }\right),\;\chi_{\phi }\left(\epsilon \right)=L^{\gamma /\nu }F_{\chi }\left(\epsilon L^{1/\nu }\right)$$
where $\epsilon \equiv \frac{\eta_{c}-\eta }{\eta_{c}}$, β and ν and γ are some exponents, and $F_{\phi }$ and $F_{\chi }$ are universal functions with the asymptotic behaviors $\lim_{x\to \infty }F_{\phi }\left(x\right)=x^{\beta }$, $\lim_{x\to \infty }F_{\chi }\left(x\right)=x^{-\gamma }$, and $\lim_{x\to 0}\left(F_{\phi }\left(x\right),F_{\chi }\left(x\right)\right)=\mathrm{constant}$. These exponents are related via a hyperscaling relation [51]
(5)νd=γ+2β,
where *d* is the Euclidean dimension of space, which is two here. In the main part of Figure 2c and its upper inset, we show the finite size dependence of ϕ in terms of η and p(ϕ) at $\eta_{c}$ for α=1.5, respectively. While the peak goes to the left as *L* increases, $L^{\beta /\nu }\phi$ is *L*-independent exactly at $\eta =\eta_{c}$ as is shown in the lower inset of Figure 2c. The values of $\eta_{c}$ obtained through this data collapse analysis agree remarkably well with those determined via the Binder cumulant method. The results of the data collapse analysis presented in Figure 2d confirm that our model obeys finite size scaling relations for sufficiently small α values in accordance with second-order phase transitions. Note that the exponents β and ν are robust against changes in α, and also $\chi^{2}$ of the fitting increases as α increases (the worse fitting is for α=1.95 in our α set), showing that the scaling hypothesis works for sufficiently small α values, and the finite-size scaling Equation (Equation 4) is not appropriate for large α values. One may use the theory of finite-size scaling for first-order transitions [47] for those cases, which is outside the scope of the present study. $\chi_{\phi }$ is reported in Figure 3a, exhibiting a pronounced peak at the transition point. $\chi_{\max}$ (the amount of $\chi_{\phi }$ at its peak) also scales with *L* by the exponent γ/ν. Figure 3b displays the α dependence of $\chi_{\phi }$ and $\chi_{\max}$, indicating that an increase in α corresponds to an increase in fluctuations of ϕ, which is the reason why increasing α favors the disordered phase. A data collapse according to Equation (Equation 4) for $\chi_{\phi }$ is presented in Figure 3c, giving γ for all α values. The γ exponent is found to be in the interval [1.4–1.6] for all α values, which agrees with the hyperscaling relation Equation (Equation 5).

The behavior of the system in terms of density is crucial in understanding the emergent collective behaviors of self-propelled particles. We found evidence regarding the density-driven order–disorder transition for super-diffusive active particles by inspecting the properties of ϕ in terms of ρ. We first considered the transition in terms of η for low density regime (ρ=0.3). In this case, similar properties, such as high density limit ρ=2, were observed, shown in Appendix A (Figure A2). We also found also a density-driven order–disorder transition, which is depicted in Figure 4, showing this transition in terms of ρ for α=1.5 and η=3. The critical density in this case was found to be $\rho_{c}=1.16\pm 0.01$, and the exponents of the transition were $\beta_{\rho }=0.16\pm 0.02$ and $\nu_{\rho }=1.32\pm 0.14$. Again, the transition showed similarities with the second-order phase transition paradigm (such as the finite size scaling, based on which we extracted the exponents using the data collapse analysis) but was not completely fitted to it.

To conclude this section, we observed that our model undergoes an order–disorder phase transition for all α values, and $\eta_{c}$ decreases as α increases. Our findings reveal that the phase transition is of the first order for large α values, similar to the ordinary Vicsek model. Conversely, when α decreases sufficiently, the transitions align better with second-order phase transitions, or at least show similarities with second-order phase transitions. We observed a crossover from first-order to second-order transition by decreasing α.

## 4. Mean Field Arguments

To comprehend the impact of α on our model’s characteristics, especially to address why the disordered phase becomes increasingly stable with increasing α, we develop a mean field theory. We consider a swarmed cluster with an average radius *r*, and study the dynamics of the particles that leave or enter the cluster as an impact of Levy flights. We quantify the rate of particle movement out of (or into) the system with $n_{\mathrm{out}}$ ($n_{\mathrm{in}}$), and define the density of the population within (or outside of) the swarmed cluster as $\rho_{\mathrm{in}}$ ($\rho_{\mathrm{out}}$). These mean field quantities are obtained through a straightforward argument illustrated in Figure 5. $n_{\mathrm{in}}$ is not sensitive to the movement of the swarmed cluster since the Levy distribution is nearly invariant under a small boost. Therefore, we use the setup in Figure 5a for determining $n_{\mathrm{in}}$ as follows (noting that on average the fraction $\frac{1}{4}$ of particles go to a required direction):(6)$$n_{\mathrm{in}}\left(r\right)=\frac{\pi \rho_{\mathrm{out}}}{2}\int_{\epsilon }^{l_{\max}}\left(r+v\right)p_{\mathrm{acu}}\left(l>v\right)dv$$
where $p_{\mathrm{acu}}\left(l>r\right)\equiv \int_{r}^{\infty }p_{\mathrm{Levy}}\left(l\right)dl$ is the Levy accumulated probability density. To calculate $n_{\mathrm{out}}$, we allow the swarmed cluster to move by a distance of ${\bar{l}}_{\alpha }$ during a single time step. The preferred direction $\langle \theta \rangle$ can be calculated by the previous step given in Figure 5b. To calculate $n_{out}$, we note that the particles in the horizontal distance $y^{\prime }$ (on the green bar) with a flight length ${\bar{l}}_{\alpha }-\left(r+y^{\prime }\right)<l<{\bar{l}}_{\alpha }+\left(r-y^{\prime }\right)$ remain in the coherent swarmed cluster. Based on this, we calculate the number of particles that leave this area. Noting that the required accumulated probability is $p_{\mathrm{acu}}\left(r,{\bar{l}}_{\alpha },y^{\prime }\right)\equiv \int_{\max\left\{{\bar{l}}_{\alpha }-\left(r+y^{\prime }\right),\epsilon \right\}}^{{\bar{l}}_{\alpha }+\left(r-y^{\prime }\right)}p_{\mathrm{Levy}}dl$, and also (noting that on average the fraction $\frac{1}{2}$ of particles go to a required direction),
(7)$$n_{\mathrm{out}}=r\rho_{\mathrm{in}}\left\{4r-\int_{-r}^{r}p_{\mathrm{acu}}\left(r,{\bar{l}}_{\alpha },y^{\prime }\right)dy^{\prime }\right\},$$
we finally find the relations
(8)$$\begin{array}{c} n_{\mathrm{in}}=\frac{\pi A\rho_{\mathrm{out}}l_{\max}^{2-\alpha }}{4\alpha }\left\{\tilde{\epsilon }\left(2x+\tilde{\epsilon }\right)+\frac{\alpha \left[(\alpha -1)-2x(2-\alpha )\right]+2{\tilde{\epsilon }}^{1-\alpha }\left[\left(2-\alpha \right)x-\left(\alpha -1\right)\tilde{\epsilon }\right]}{\left(2-\alpha )(\alpha -1\right)}\right\}\\ n_{\mathrm{out}}=r\rho_{\mathrm{in}}\left\{4r-\frac{A}{\alpha }\left[\frac{1}{\alpha -1}\left(\epsilon^{1-\alpha }+{\left({\bar{l}}_{\alpha }+2r\right)}^{1-\alpha }-2{\bar{l}}_{\alpha }^{1-\alpha }\right)+\epsilon^{-\alpha }\left(2r-{\bar{l}}_{\alpha }\right)\right]\right\}, \end{array}$$
where $x\equiv r/l_{\max}$ and $\tilde{\epsilon }\equiv \epsilon /l_{\max}$. To obtain the mean field relations for $\rho_{\mathrm{in}}$ and $\rho_{\mathrm{out}}$, we use the following argument: In the ordered phase, there are typically multiple particles inside a disk with area $\pi R^{2}$, whereas in the disordered phase, particles are unlikely to encounter each other within a single Levy flight, resulting in no particles within an area of ${\bar{l}}_{\alpha }R$. This leads to the relation $\rho_{\mathrm{in}}\approx \left(\frac{{\bar{l}}_{\alpha }}{\pi R}\right)\rho_{\mathrm{out}}$. It is worth noting that the relationship is not qualitatively dependent on the power used in the equation. This, along with conservation of the total number of particles $r^{2}\rho_{\mathrm{in}}+\left(l_{\max}^{2}-r^{2}\right)\rho_{\mathrm{out}}=\rho l_{\max}^{2}$, gives the following relation:(9)$$\rho_{\mathrm{in}}=\frac{\rho }{\left(1-\frac{\pi R}{{\bar{l}}_{\alpha }}\right)x^{2}+\frac{\pi R}{{\bar{l}}_{\alpha }}},\;\rho_{\mathrm{out}}=\frac{\frac{\pi R}{{\bar{l}}_{\alpha }}\rho }{\left(1-\frac{\pi R}{{\bar{l}}_{\alpha }}\right)x^{2}+\frac{\pi R}{{\bar{l}}_{\alpha }}}.$$
Noting that $n\equiv n_{\mathrm{in}}-n_{\mathrm{out}}$ is the rate of change of the average number of active particles inside the swarmed cluster, one can determine the dynamical behaviors of the model in terms of *r*. The average *r* (which we call $r^{*}$) is the fixed point of the dynamical behavior of *n*, i.e., $n(r^{*})=0$. The average size of the swarmed clusters, indicated by $r^{*}$, is used to determine the stability of each phase for a given value of α.

The relationship between the average size of the swarmed clusters and the parameter α is presented in Figure 6. The figure shows that as α increases, the disordered phase becomes more stable.

To make connection between $r^{*}$ and $\eta_{c}$, we first note that $r^{*}\to 0\Rightarrow \eta_{c}\to 0$ and $r^{*}\to \infty \Rightarrow \eta_{c}\to \infty$. Therefore, we expect that $\eta_{c}$ is a decreasing function of α, which agrees with the simulation results. As pointed out in [52] for a standard Vicsek model with large densities, the system may percolate and form a huge cluster. This happens when $r^{*}\to \infty$ in our MF arguments. For the percolation probability, we should seek the conditions that lead to $r^{*}\to \infty$, or equivalently $r^{*}/L\to 1$. In our model, the probability of percolation is greater for the smaller α values. Given that in this limit, the fluctuations are much greater than for large α values (because of the criticality of the system at this limit), it is consistent with the percolation theory which tells us that at the percolation threshold, that fluctuations are maximal. In the inset of Figure 6, we show the relation between $\eta_{c}$ and $r^{*}$, which is monotonic increasing function. If one fits this relation using a power-law function, the exponent would be ≈0.07, although the range of the quantities are too small to deduce a power-law form (it commonly should be more than one decade).

## 5. Geometrical Observables

In addition to local characteristics, systems undergoing continuous phase transitions display global geometric features, which have been the subject of numerous analytical and simulation studies. This characterization can reveal previously hidden aspects of the models that are not apparent with only local observables. Section 5.1 focuses on the fractal analysis of the density of active particles at the transition point. In Section 5.2, the critical loop ensemble (CLE) of the iso-density loops on the system is analyzed.

### 5.1. Density Fractal Analysis

We employ the fractal analysis method for the density configurations [53] that we obtain at the transition points. The density configurations are first converted into black-and-white images, after which the white pixels are statistically analyzed; see Figure 1. The system is meshed using boxes of a specific size linear size δ, and the statistics of the *filling fraction* of each box is calculated. A pixel is considered white or occupied if the density of active particles at that site ρ is greater than the spatial average of density $\bar{\rho }\equiv N_{\mathrm{pixels}}^{-1}\sum_{i}\rho_{i}$ over that sample, where $N_{\mathrm{pixels}}$ is the total number of pixels in the system. The filling fraction of the *i*th box is $\mu_{i}\equiv \frac{N_{i}\left(\delta \right)}{N_{\mathrm{pixels}}}$, where $N_{i}\left(\delta \right)$ is the number of white pixels in the *i*th box [54,55]. Note that $\sum_{i=1}^{N_{\mathrm{box}}}N_{i}=N_{\mathrm{pixels}}$, where $N_{\mathrm{box}}$ is the total number of boxes. If we attribute a local *mass* for *i*th box as $m_{i}\left(\delta \right)\equiv 1-\delta_{N_{i}\left(\delta \right),0}$ and a *total mass* to the cluster as $M\left(\delta \right)\equiv \sum_{i}m_{i}\left(\delta \right)$, where $\delta_{m,n}$ is a Kronecker delta, then the box counting fractal dimension is obtained as
(10)$$D_{f}\equiv -\lim_{\delta \to 0}\frac{\log M(\delta )}{\log\delta }.$$

In a multifractal system, this exponent depends on the scale that we are considering or changes from region to region. A unified standard theory called multifractal analysis was previously developed, which employs a generalized partition function that yields a spectrum of exponents, including the fractal, information, and correlation dimensions [53]. This *q*-generalized partition function is related to the *q*th moment of the fluctuations of $\mu_{i}$, and is defined as
(11)$$Z_{q}\left(\delta \right)=\sum_{i}{\left[\mu_{i}\left(\delta \right)\right]}^{q},$$
where *q* is a moment. For scale-invariant systems, $Z_{q}$ scales with δ in a power-law form with the exponent $\gamma_{q}$, but the exponent may not be a unique number in all scales:(12)$$Z_{q}\left(\delta \right)\propto \delta^{\gamma_{q}},\;\mathrm{so}\;\mathrm{that}\;\gamma_{q}=\lim_{\delta \to 0}\frac{\log Z_{q}\left(\delta \right)}{\log\delta }.$$The generalized *q*-dimension is then defined as
(13)$$D_{q}\equiv \frac{\gamma_{q}}{q-1},$$
so that $D_{f}=\lim_{q\to 0}D_{q}$. Note that if one interprets $\mu_{i}$ as a probability associated with a small segment (δ) of the system, then $D_{q}$ plays the role of a normalized *q*-Renyi entropy ($Re_{q}\left(\delta \right)$) in the thermodynamic limit δ→0
(14)$$Re_{q}\left(\delta \right)\equiv \frac{1}{1-q}\log\sum_{i}{\left[\mu_{i}\left(\delta \right)\right]}^{q},$$
so that
(15)$$D_{q}=-\lim_{\delta \to 0}\frac{Re_{q}\left(\delta \right)}{\log\delta }.$$Therefore, the mass fractal dimension of samples is related to q=0 Renyi entropy
(16)$${\left.Re_{q=0}\left(\delta \right)\right|}_{\delta \to 0}=-D_{f}\log\delta .$$
It is worth noting that the hypothesis of scale invariance, as described in Equation (Equation 12), has led to the fact that Renyi entropy is proportional to logδ and not $\delta^{d}$ (d=2 in our case) as expected for the ordinary systems. This serves as an important characteristic of the scale-invariant systems, for which the system is not extensive [56]. The information dimension associated with the Shannon entropy is obtained in the limit q→1
(17)$$\mathrm{SH}\left(\delta \right)\equiv -\sum_{i}\mu_{i}\log\mu_{i},$$
the fact that relates it to $D_{1}$
(18)$$D_{1}\equiv \lim_{\delta \to 0}\frac{\sum_{i}\mu_{i}\left(\delta \right)\log\mu_{i}\left(\delta \right)}{\log\delta }=\lim_{q\to 1}D_{q},$$
so that
(19)$${\left.\mathrm{SH}(\delta )\right|}_{\delta \to 0}=-D_{1}\log\delta .$$Finally the correlation dimension is defined as
(20)$$C\equiv \lim_{\delta \to 0}\frac{\log C(\delta )}{\log\delta },$$
where
(21)$$C\left(\delta \right)\equiv \frac{1}{N_{\mathrm{pixels}}^{2}}\sum_{k\neq k^{\prime }}\Theta \left(\delta -\left|R_{k}-R_{k^{\prime }}\right|\right),$$
where $R_{k}$ is the position of the *k*th white pixel (not box), and Θ is a step function. It is shown that [54]
(22)$$C=D_{2}.$$

We generated $10^{3}$ configurations (density snapshots) at the transition points to investigate the anomalous dimensions. Figure 7a shows $\log Z_{q}\left(\delta \right)$ in terms of log(δ) for α=1.5, and the inset shows $\gamma_{q}$ in terms of *q*, which is well described by a linear function. The numerical values for the dimensions are reported in the inset of Figure 7b. For the small α values, the exponent remains constant and stable across various α values. Interestingly, the mass fractal dimensions are lower than one, which is generally possible for the fractals with fractional filling boxes. However, as α approaches two, power-law fittings fail to fit, and the resulting exponents deviate from the others. This is not surprising since, at these points, the system does not display a fractal structure.

### 5.2. Contour Line Analysis

The critical (conformal) loop ensemble (CLE) theory enabled another type of classification of two-dimensional (2D) critical models based on their global geometrical properties [57,58]. When considering this approach, the focus is on random curves that can be transformed into dynamic stochastic paths, or exploration processes, within a connected domain in the plane. This idea was originally suggested by Loewner and is known as stochastic Loewner evolution (SLE) [59,60], which is now widely recognized as a means of characterizing the interfaces of two-dimensional statistical models using growth processes. The method can classify these interfaces into one-parameter classes, with the diffusivity parameter κ serving as the representative parameter, called SLE_*κ*_ [59,60]. Many other exponents are related to κ, such as the fractal dimension of level lines or interfaces, which is [61] $d_{f}=1+\frac{\kappa }{2}$ (these extended objects are fractal paths, or sometimes loops). As another example for the Potts models, it is shown that $\nu \left(\kappa \right)=\frac{2}{3(2-\bar{\kappa })}$, $\gamma \left(\kappa \right)=\frac{4+3{\bar{\kappa }}^{2}}{6\bar{\kappa }\left(2-\bar{\kappa }\right)}$ and $\beta \left(\kappa \right)=\frac{3\bar{\kappa }-2}{12\bar{\kappa }}$, where $\bar{\kappa }\equiv 1/\kappa$ is the dual of κ, which is given by the equation $\sqrt{Q}=-2\cos\left(\pi /\bar{\kappa }\right)$ such that, for example, $\bar{\kappa }=\frac{4}{3}$ for Q=2 [62].

To investigate this, we start with the fractal dimension of loops ($d_{f}^{\mathrm{loop}}$) for *a single connected cluster*, which is defined as the scaling exponent between the loop length *l* and the loop gyration radius *r*. The latter is defined for a closed path $\left\{{\vec{r}}_{1},{\vec{r}}_{2},\ldots ,{\vec{r}}_{l}\right\}$ as $r^{2}\equiv \frac{1}{l}\sum_{k=1}^{l}{\left|{\vec{r}}_{k}-{\vec{r}}_{\mathrm{com}}\right|}^{2}$, where ${\vec{r}}_{\mathrm{com}}\equiv \frac{1}{l}\sum_{k=1}^{l}{\vec{r}}_{k}$ is the loop center-of-mass. $d_{f}^{\mathrm{loop}}$ is then defined by the relation $\langle \log(l)\rangle =d_{f}\langle \log(r)\rangle +\mathrm{cont}.$, where 〈〉 denotes the ensemble average. We apply the Hoshen–Kopelman algorithm [63] to identify the connected components of the clusters. This algorithm involves coloring the entire cluster and assigning different colors to separate clusters while traversing the sample. We analyze these connected clusters by measuring their external boundaries with length *l*, as well as their gyration radius *r* and mass, which we denote as sm. The scale-invariant properties of the distribution function of sm, *l*, and *r* are also evident (excluding finite size effects), with $P\left(x\right)\propto x^{-\tau_{x}}$. Here, $\tau_{x}$ represents the corresponding scaling exponent.

We consider clusters that are associated with the white-and-black density pattern. The scaling hypothesis in the transition points is supported by the findings presented in Figure 8a,b. It should be noted that when the value of α approaches two, the system loses its scale-invariance property and also becomes anisotropic, indicating that the critical or scaling exponents cannot be considered dependable in this range. Given the system and the associated exponents, the fractal dimension $d_{f}^{\mathrm{loop}}$ is observed to remain constant at approximately 1.40 for all values of α, indicating its robustness. This exponent is associated with κ=0.8 (in the case of conformal invariance). However, the scaling exponents $\tau_{x}$ (x≡sm,l,r) exhibit changes as α varies. This is the first time in this study that we see the set of exponents change as a function of α, suggesting that a range of systems can be visited within this interval of α. It would be intriguing to compare the results with the ones for the geometric and Fortuin–Kasteleyn (FK) clusters of the critical Q=2 Potts (Ising) model, the diffusivity parameters, which are $\kappa_{G}=\frac{3}{4}$ and $\kappa_{\mathrm{FK}}\equiv {\bar{\kappa }}_{G}=1/\kappa_{G}=\frac{4}{3}$, respectively [62]. It also was previously shown that $\tau_{r}^{\mathrm{Ising}}\approx 3.4$, and $\tau_{l}^{\mathrm{Ising}}=\frac{d}{D_{f}}+1\approx 2.5$, and also $d_{f}^{\mathrm{Geometrical}}=1+\frac{3}{8}=1.375$ [64]. The exponents that we found for our model for small α values are in agreement with the exponents explored above. By setting κ=0.8 in our model, we obtain $d_{f}\left(\kappa \right)=1.4$, ν(κ)≈0.9, γ(κ)≈1.55 and β(κ)≈0.12. These values are consistent with our simulation results, except for the β exponent, which shows a discrepancy. This suggests that the interfaces of our model do not exhibit conformal invariance. Hence, we conclude that our model is self-similar for sufficiently small α values, and exhibits some similarity to the Q=2 Potts model but is not a perfect fit for this model.

## 6. Concluding Remarks

In this paper, we studied a super-diffusive variant of the Vicsek model by introducing scale-free (Levy) stochasticity to the flights taken by the active particles during each time step. As a result of this modification, the transition, which is of the first order for the conventional Vicsek model, shows similarities with second-order phase transitions for small α values. Since we observed some features of first-order phase transitions, we denote this regime as weakly second-order phase transitions. In contrast, for α values around two, our model displays first-order phase transitions, but like the Vicsek model, it also exhibits some characteristics of scale invariance. The latter led Vicsek et al. to incorrectly conclude that their model was of the second order and calculate “fictitious exponents” [11]. This occurs when two peaks of p(ϕ) are in close proximity and difficult to distinguish as shown in the right inset of Figure 2a. Since the values of $\eta_{c}$ and the “fictitious exponents” for our model (for α values close to two) are similar to those of the ordinary Vicsek model, we can infer that our model exhibits the same characteristics as the ordinary Vicsek model at the transition point for α values close to two. Additionally, for α≥2, the α-stable Levy systems are unstable towards a fixed point that is characterized by a Gaussian distribution function [44], indicating that perturbing the ordinary Vicsek model with Gaussian-distributed flights does not alter the fundamental properties of the Vicsek model, and is therefore an “irrelevant perturbation”, while for the small α values, it is similar to second-order phase transitions and the corresponding perturbation is relevant. We observe a crossover from first-order phase transition (large α values) to weakly second-order transitions (small α values).

We developed a mean field theory for our model, which successfully describes why the disordered phase becomes more stable as α increases. The geometrical properties of the model at the transition points were also investigated. We found a series of anomalous dimensions, including the mass dimension, the information dimension and the correlation dimension. Our critical loop ensemble study shows that this system has similarities to the Q=2 Pottes (Ising) model, while the β exponent does not match.

## Figures and Tables

**Figure 1 entropy-25-00817-f001:**
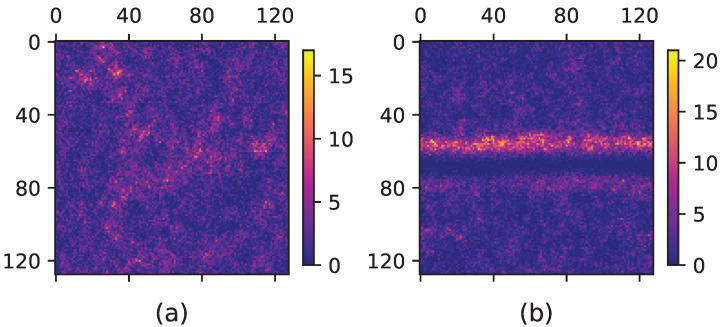
A snapshot of the particle density in the ordinary and super-diffusive VM (our model) at the transition point. A color map for α=1.5 and α=1.95 is shown in (**a**) and (**b**).

**Figure 2 entropy-25-00817-f002:**
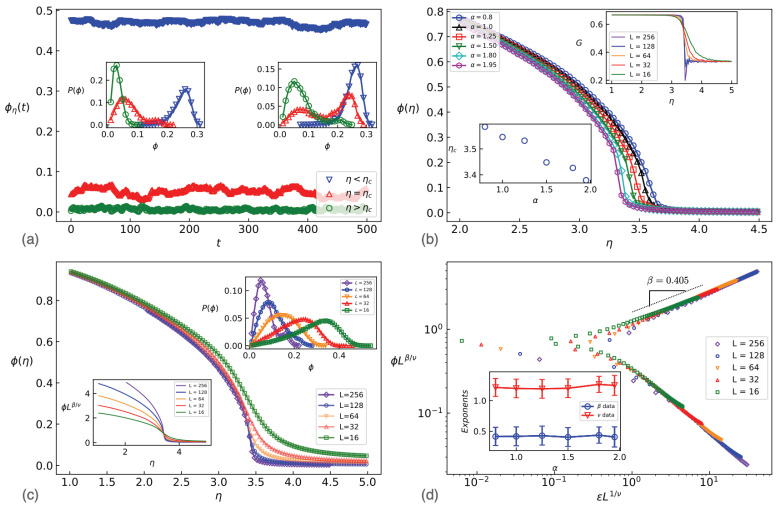
(**a**) The time series of ϕ for various amounts of η for L=256. Left (right) inset shows the probability distribution of ϕ(η) for $\eta <\eta_{c}$, $\eta =\eta_{c}$ and $\eta >\eta_{c}$ for α=1.5 (α=1.95). (**b**) ϕ-η graph for various α values, showing the transition structure. Upper inset shows that Binder cumulant *G* in terms of η for α=1.5, which gives the transition point as its coincidence point. Lower inset shows the transition point in terms of α. (**c**) ϕ-η graph for various *L* values. The coincidence point of re-scaled ϕ is shown in the lower inset in terms of η, which determines the transition point. The upper inset shows how the peak of the distribution functions runs with the system size. (**d**) $\log(L^{\beta /\nu }\phi \left(\eta \right))$ in terms of $\log\epsilon L^{1/\nu }$, exhibiting a scaling behavior according to Equation (Equation 4). The inset shows the exponents in terms of α.

**Figure 3 entropy-25-00817-f003:**
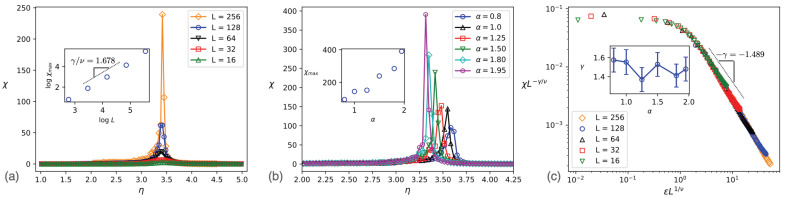
χ in terms of η, exhibiting a divergent behavior at the transition point (**a**) for various system sizes and α=1.5 and (**b**) various α values and L=128. The inset of (**a**) is $\log\chi_{max}$ in terms of logL the slope of which is γ/ν, and the inset of (**b**) is $\chi_{max}$ in terms of α. (**c**) re-scaled χ in terms of re-scaled η with the slope −γ.

**Figure 4 entropy-25-00817-f004:**
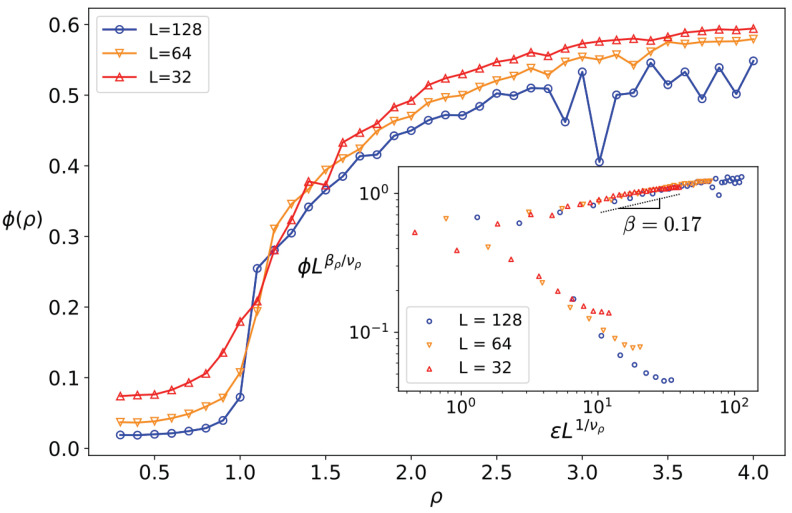
ϕ in terms of ρ for η=3 and α=1.5 for various *L* values (up to L=128). Inset: The data collapse analysis for ϕ in terms of ρ for η=3, where the reduced density is defined as $\tilde{\rho }\equiv \frac{\rho_{c}-\rho }{\rho_{c}}$.

**Figure 5 entropy-25-00817-f005:**
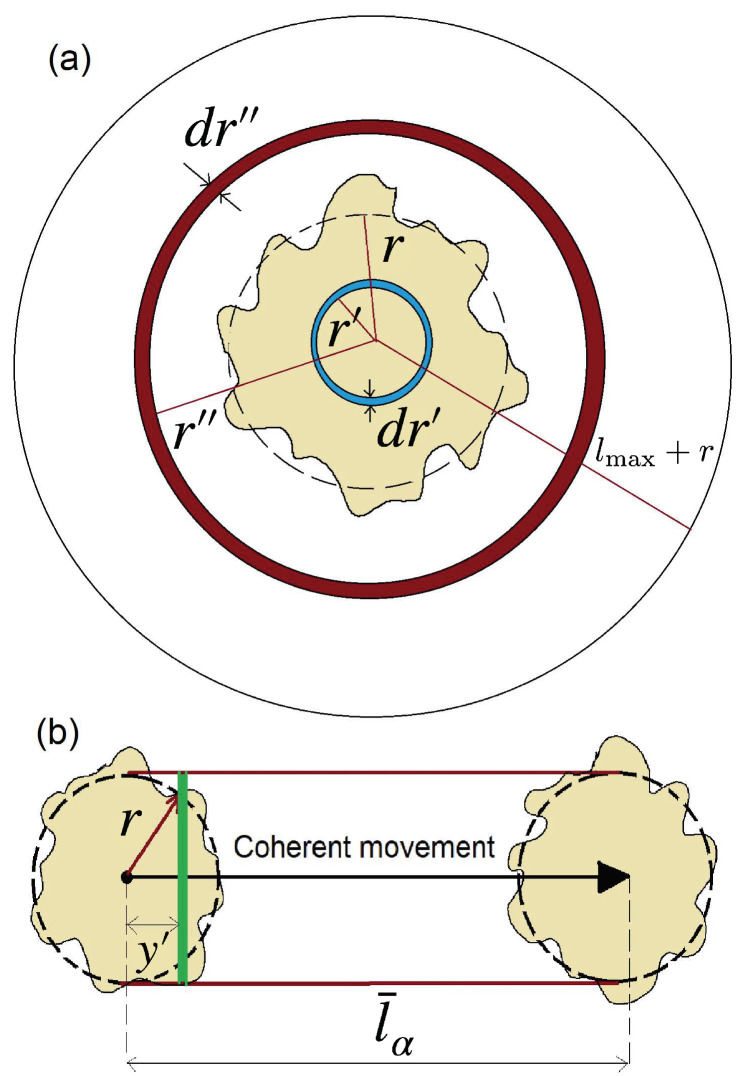
Schematic representation of the mean field method. (**a**) A static swarmed cluster (the yellow area shows), where *r* is its average radius, and the red and blue rings indicate the number of particles entering $n_{in}$ and leaving $n_{out}$ the cluster, respectively. (**b**) The coherent movement of the swarmed cluster in the preferred direction $\langle \theta \rangle$. The green bar moves from $y^{\prime }=-r$ to $y^{\prime }=r$ running over the area inside the swarmed cluster. The active particles with Levy flights in the range $\left[{\bar{l}}_{\alpha }-\left(r+y^{\prime }\right),{\bar{l}}_{\alpha }+\left(r-y^{\prime }\right)\right]$ remain inside the swarmed cluster if the average radius *r* remains approximately unchanged during the process. The number of such particles is $N-n_{\mathrm{out}}$, where *N* is the number of particles inside the cluster in the previous step.

**Figure 6 entropy-25-00817-f006:**
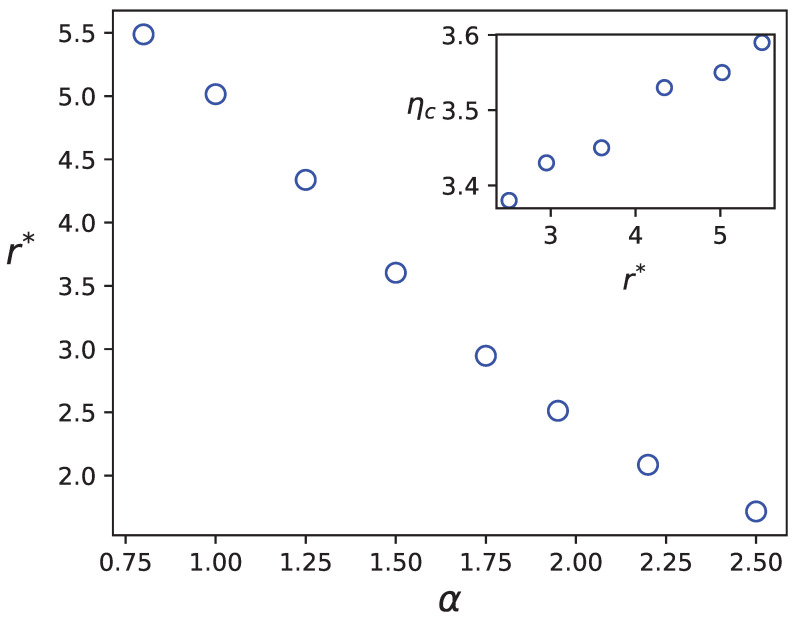
$r^{*}$ in terms of α based on the mean field results.

**Figure 7 entropy-25-00817-f007:**
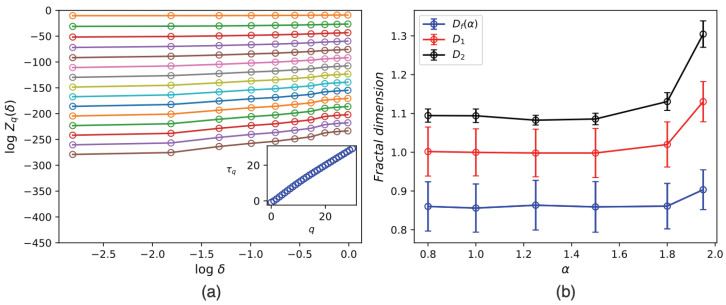
(**a**) $\log Z_{q}$ in terms of logδ for q∈[0,30] in increment 2 for α=1.5 (from the top to the bottom *q* decreases), the slope of which is γ(q) (inset). (**b**) The fractal dimension ($D_{f}$), the information dimension ($D_{1}$) and the correlation dimension $D_{2}$ in terms of α.

**Figure 8 entropy-25-00817-f008:**
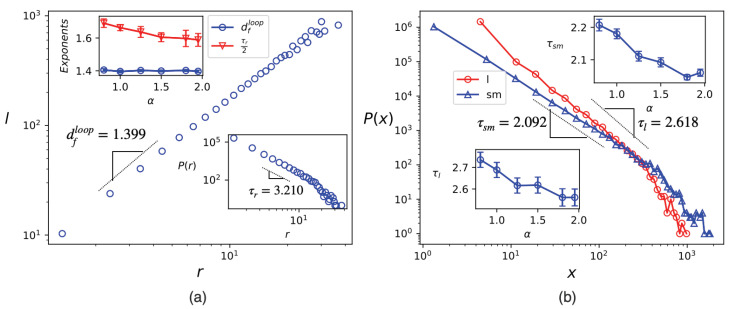
(**a**) logl in terms of logr for α=1.5, the slope of which is $d_{f}^{\mathrm{loop}}$. Lower inset: the distribution of the gyration radius *r* for α=1.5. Top inset: $d_{f}^{\mathrm{loop}}$ and $\tau_{r}/2$ in terms of α. (**b**) The distribution of the loop length (x=l) and submass (x=sm). Top (Down) inset shows $\tau_{sm}$ ($\tau_{l}$) in terms of α.

## Data Availability

The research data supporting this publication are provided within this paper.

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
