# Peer review of "Swarming Transition in Super-Diffusive Self-Propelled Particles"

_entropy, 2023, doi:10.3390/e25050817_

Round 1

Reviewer 1 Report

The draft entitled “Swarming Transition in Super-Diffusive Self-Propelled Particles” by Morteza Nattagh Najafi et.al has been reviewed. The paper extends the Vicsek model by incorporating the long-range movement (i.e. Levy flights). They find that in a region of the parameter \alpha, the swarming swarming transition is of second order, while the transition is first-order in the region. They develop a mean field treatment to explain this. 

Overall, the observations are reasonable and in line with previous studies. Regarding the debate of nature of phase transition in the classic Vicsek model, it’s generally believed that as long as the particle interaction is short-ranged, the discontinuous phase transition is always expected. But, if there are long-ranged particle interactions, continuous phase transitions are expected instead. With these consensus, the observations of second order phase transition are reasonable, because the super-diffusion introduces some long-range moments/interactions.

Even though, I still think this work is still a contribution to the field of swarming transition, as the Levy flight (or super-diffusion) is seen in many species; therefore, including this property into the swarming dynamics is quite natural. Though, I have some minor comments for their improvement.

As stated in the above, the introduction of long-ranged interactions can make the continuous transition was ever discussed in the community. But I cannot remember the specific references. So if possible, please find the related references to enhance the arguments of this observation. 

The question raised “how does the criticality enter in the swarmed active systems (as already observed [9]), and what are its impacts.” seems not a  

good motivation for this study, or so to say, too theoretic to give a good impression. In fact, I believe there are many ways to make the swarming phase transition to be continuous, the introduction of Levy flight is only one of them. So the current study leads to criticality, but it’s hard to symmetrically address how criticality enters. Therefore, I suggest to change the motivation from the more empirical aspect, e.g. given the Levy flight is so widely seen, it’s natural to see how this observation impacts the collective motion when these individuals are swarming together.

There are two some minor places: 

 Page 3, Second column, 5 line, the italic word “contentious” seems to be “continuous”; 

The wording \phi-\eta graph, “graph” in nowadays has some specific meaning, better to be replaced by plot, dependence, phase transition etc.

With these, the reviewer think the submission overall makes a solid contribution to this field, it can be accepted in Entropy after a minor revision by considering the issues raised above.

Reviewer 2 Report

The paper explored the phase behavior of the Vicsek model incorporated with Levy flight with an exponent α. The authors analyzed the critical behaviors of the system with different α. They argued that a ‘critical’ α, i.e., α_tcr, exists. Below α_tcr, the phase transition of the system is of second order, while upon this ‘critical’ alpha, the phase transition is of the first order. To further understand these results, they proposed a mean-field theory based on the growth of the swarmed clusters, from which they obtained different critical exponents consistent with their simulation results. The obtained geometry exponents revealed the similarities of the geometric properties between the 2D Q = 2 Potts (Ising) model and the model used in the paper.

The presence of scale-free stochasticity is widely observed in experiments and nature. Meanwhile, using simulations and mean-field arguments, the authors provide some new features of the Vicsek model with the velocity that obeys Levy flight distribution. It is also interesting to compare their model and 2D Q=2 Potts model from geometric properties. I recommend the paper for publication once the authors satisfactorily address the points raised below. 

1. For the standard Vicsek model, the crossover size to show a first-order signature is typically very large. Therefore, my main concern is that although the paper argued that the phase transition of the present system with α is away from 2 is of second order. From the data provided, I cannot see that it rules out the possibility that the phase transition for small α is still of first order when system size becomes large. 

Especially, as it is known, a sharp minimum of Binder cumulant toward negative values is one of the signatures for a first-order-like transition. In the inset of Fig. 2 (b), the Binder cumulant for α=1.5 and L=128 shows the tendency to have a sharp minimum around ‘η_c’. If the system size increases, for instance, say L=256, will the minimum value of Binder cumulant jump into negative? 

Besides, in Sec. III, the authors mentioned about the hysteresis effect. Do they have any results to show the hysteresis? 

2. The authors provided a mean-field argument to discuss the phase transition in the system based on the swarmed clusters. As mentioned in ‘PHYSICAL REVIEW E 100, 022606 (2019)’ that for a standard Vicsek model with large densities (e.g., ρ >= 2), the system may percolate and form a huge cluster. Will a similar story also exist in the present model? How will these results affect the mean-field argument?

3. The authors only considered one density in the paper. What will happen when the system becomes dilute? For instance, when ρ=0.3, will the phase behavior changes? Or is there also a critical α?  

4. Minor points: 

-- In Sec. V, the first two paragraphs are almost the same. The authors should delete one of them.

-- Fig. 1 provided colorful and black-and-white snapshots for the same α, which is unnecessary. I suggest keeping the colorful snapshots.

-- In Sec. III, the authors says that they simulate the system with L=32, 64, 128, and 256, however, in the figures, the system sizes are within 16 to 128.

See above

Round 2

Reviewer 1 Report

Now the authors have addressed all concerns raised in the last round. I am happy to recommend its publication in Entropy.

Reviewer 2 Report

The authors have properly addressed my concerns. They soften their main claim about the second-order phase transition for small α, provide new simulation results of the Binder cumulant, and discuss the diluted conditions of the system. They also rewrite the paper well. Therefore, I recommend publishing the paper in its current form.